# Reduced Hippocampal Volumes in Children with History of Hypoxic Ischemic Encephalopathy after Therapeutic Hypothermia

**DOI:** 10.3390/children10061005

**Published:** 2023-06-02

**Authors:** Katie M. Pfister, Sally M. Stoyell, Zachary R. Miller, Ruskin H. Hunt, Elizabeth P. Zorn, Kathleen M. Thomas

**Affiliations:** 1Department of Pediatrics, University of Minnesota, 2450 Riverside Ave., AO-401, Minneapolis, MN 55454, USA; zornx011@umn.edu; 2Institute of Child Development, University of Minnesota, Campbell Hall, 51 E River Rd., Minneapolis, MN 55455, USA; stoye003@umn.edu (S.M.S.); mill5076@umn.edu (Z.R.M.); thoma114@umn.edu (K.M.T.)

**Keywords:** hypoxic ischemic encephalopathy, neurodevelopment, memory function, MRI, hippocampus

## Abstract

Hypoxic ischemic encephalopathy (HIE) remains a significant cause of disability despite treatment with therapeutic hypothermia (TH). Many survive with more subtle deficits that affect daily functioning and school performance. We have previously shown an early indication of hippocampal changes in infants with HIE despite TH. The aim of this study was to evaluate the hippocampal volume via MRI and memory function at 5 years of age. A cohort of children followed from birth returned for a 5-year follow-up (*n* = 10 HIE treated with TH, *n* = 8 healthy controls). The children underwent brain MRI and neurodevelopmental testing to assess their brain volume, general development, and memory function. Children with HIE had smaller hippocampal volumes than the controls despite no differences in the total brain volume (*p* = 0.02). Children with HIE generally scored within the average range on developmental testing. Though there was no difference in the memory scores between these groups, there was a positive within-group correlation between the hippocampal volume and memory scores in children with HIE (sentence recall r = 0.66, *p* = 0.038). There was no relationship between newborn memory function and 5-year hippocampal size. Children with HIE treated with TH experienced significant and lasting changes to the hippocampus despite improvements in survival and severe disability. Future studies should target diminishing injury to the hippocampus to improve overall outcomes.

## 1. Introduction

Hypoxic ischemic encephalopathy (HIE) after birth affects around 1–8/1000 live births [1]. The impact can be severe in infants with moderate or severe encephalopathy, with nearly half of infants impacted by death or moderate to severe disability [2,3]. Though therapeutic hypothermia (TH) has reduced the incidence of death and severe disability, there continues to be a high prevalence of subtle cognitive impairments in children without severe deficits or cerebral palsy [3,4,5]. Furthermore, it is now recognized that even infants with mild encephalopathy have lower cognitive scores than healthy controls [6].

Memory function, which affects school performance and daily functioning, is negatively impacted by HIE [7,8]. Prior to the advent of TH, studies have shown that children with a history of HIE had poorer working memory, long-term episodic memory, and verbal and visual memory [9,10,11]. While some studies have shown continued memory dysfunctions in children with history of HIE even after TH [3,12], our group showed that *newborns* with HIE who received TH had preservation of recognition memory function at an electrophysiologic level, yet an alteration in the brain circuitry underlying that function at 2 weeks of age [13].

Previous studies using childhood MRIs have shown decreased hippocampal volumes after HIE, with associated differences in memory function at school age. However, nearly all of these studies have been completed in children who did not receive TH [8,10]. Furthermore, the results are often confounded by including preterm infants or subjects who experienced a hypoxic event later in infancy [7,14,15,16], both of which affect the stage of brain development and potentially the resultant effect of the insult [17]. Very few MRI studies have looked at a homogenous group of term-born children with a history of HIE at birth. 

In the present study, we followed a cohort of children initially recruited after birth who received TH for HIE. This cohort has been followed at several time points thus far. At 2 weeks of life, they underwent memory function testing using event-related potentials (ERP). ERPs are tiny segments of EEG that coincide with a stimulus (auditory in this case) and reflect the brain’s processing of the stimulus. At 2 weeks, infants with HIE showed discrimination between their mother’s voice compared to a stranger’s voice, similar to healthy controls, but predominantly at the midline instead of the left hemisphere [13]. At 8 months, we used ERP to evaluate the speed of processing of a visual stimulus, and at 12 months, they underwent standardized developmental and memory testing. Briefly, findings were similar to healthy controls at each of those time points [18].

The aim of this study was to evaluate brain volumes and the association between hippocampal volume and memory function at 5 years of age in this cohort of term infants affected by HIE and treated with TH. We hypothesized that memory function would be similar to the healthy controls given our early findings of an intact discrimination memory at 2 weeks after HIE [13], and this would be reflected in hippocampal volumes similar to the controls. 

## 2. Materials and Methods

### 2.1. Enrollment

This is a longitudinal cohort study. Participants for this follow-up study were recruited from a previously studied cohort of infants that included 2 groups: a group of infants with neonatal HIE, as well as a comparison group of normal, healthy control infants [13]. This 5-year follow-up study took place from 2017 to 2018. Briefly, the study cohort included a group of infants treated for HIE with TH per unit protocol in the neonatal intensive care units of the University of Minnesota Masonic Children’s Hospital, Children’s Minnesota—St. Paul campus, or North Memorial Medical Center, and were discharged by 4 weeks of age. (TH criteria were similar to published guidelines during that time period based on metabolic acidosis and an exam significant for moderate or severe encephalopathy [19], though three infants with mild HIE received TH per attending discretion). A group of healthy controls (5 min Apgar ≥ 7 and no health concerns) were recruited from the newborn nursery at the University of Minnesota Medical Center during the same time period. All participants were ≥36 weeks of gestational age at birth. The exclusion criteria for both groups at initial recruitment were the presence of a congenital abnormality or metabolic disease that would affect neurodevelopment, birth weight that was small for their gestational age, or referral on the newborn hearing screen conducted per institution guidelines. We aimed to evaluate the same group of children with brain MRI and neurodevelopmental assessment when they were 5 years old, and all participants that completed the study in the initial cohort were invited to participate in this follow-up study. The Institutional Review Board at the University of Minnesota approved the study, parents provided written consent, and children provided assent for participation. 

### 2.2. Developmental Testing

Participants underwent psychometric evaluation by one of two trained graduate students using three standardized assessments (they were blinded to the child’s newborn history and MRI results). General development and full-scale intelligence quotient (FSIQ) were assessed using the Weschler Preschool & Primary Scale of Intelligence (WPPSI) (population average composite score 100 ± 15). Memory and executive function were assessed via selected sub-scales of the NEPSY-II (subsections used: inhibition (for those children ≥5 years old), comprehension of instructions, narrative memory, sentence repetition) (population average scaled score 10 ± 2) and the Behavior Rating Inventory of Executive Function-Preschool (BRIEF-P) (population average T-scores 50 ± 10),which was completed by parents. Of note, for BRIEF-P scoring, scores higher than the average range signified a worse executive function: T-score 60–64 was mildly elevated, and a score of 65 or higher was significantly elevated.

### 2.3. MRI Acquisition and Processing 

Participants underwent an age-appropriate scan simulation using a play tunnel, prism glasses, toy masks, and recorded scanner sounds. During the MRI scan, the Siemens full headphones were used to provide padding and audio instead of traditional foam padding.

Magnetic resonance images were collected using a Siemens 3 Tesla Prisma scanner with a 32-channel head coil. A T1-weighted magnetization prepared the rapid gradient echo (MPRAGE) scan and acquired three-dimensional structural image (TR = 2530.0 ms, TE = 3.65 ms, FOV = 256 × 176 mm, flip angle = 7°, 1 mm^3^ isotropic voxel, 240 slices). The T1 image took 12 min, and children who were able to continue stayed for 7 min of DTI imaging (not enough data to include in the analysis);total time in the scanner was 30 min on average. These images were intensity normalized, and whole brain segmentation, parcellation, and volume calculation of normalized images were completed using the recon-all function of Freesurfer image analysis suite version 6.0.0 (https://surfer.nmr.mgh.harvard.edu (accessed on 7 October 2020). Hippocampal segmentation was completed using hippocampal subfields processing included in Freesurfer image analysis suite version 7.1.1 [20].

Trained staff and a graduate student inspected the images and Freesurfer output for structural anomalies, image quality, and segmentation accuracy. The MRIs were also reviewed by a neuroradiologist for incidental clinically significant findings (i.e., Structural anomalies or pathologies), and none were reported. 

### 2.4. Statistical Analysis

Due to the small sample size resulting in non-normal distributions, non-parametric tests were used for all analyses. The differences between groups in demographic variables, hippocampal volumes, and neuropsychological assessments were tested either with a Kruskal–Wallis test for continuous outcomes or with a Fisher’s exact test for categorical outcomes. The relationships between hippocampal volumes and neuropsychological outcomes were tested using Spearman’s rank-order correlation. We did not control for multiple comparisons. All analyses were performed using R Statistical Software (version 4.0.3, R Foundation for Statistical Computing, Vienna, Austria). Statistical significance was defined as *p* < 0.05.

## 3. Results

### 3.1. Study Cohort

Parental consent for participation was obtained from 26 subjects in this follow-up study (12 from the group with HIE and 14 from the control group). Of the original 48 newborn participants, 5 were lost to follow-up or dropped out during the initial 12-month infant study, and 17 were unable to be contacted or declined participation at 5 years, leaving 26 participants at 5 years. Adequate MRI scans and developmental testing were obtained in 10/12 children with a history of HIE (two participants without MRI data: one with motion artifact, one child refused MRI) and 8/14 controls (six participants without MRI data due to parent/participant consenting/assenting to developmental testing only) who participated at 5 years (Figure 1, Flowchart of participation). “Adequate” MRI was defined as structural MRI data that passed the Freesurfer segmentation process and a visual inspection by research staff. There was no statistically significant difference in the participant characteristics between those who returned for a 5-year follow-up and those who did not, both overall and within each group. Though motor function was not formally assessed, all returning participants were ambulatory and without a reported diagnosis of cerebral palsy. All participants with HIE had been referred to the state’s early intervention program and the NICU Follow-Up Clinic associated with their NICU at the time of hospital discharge. We did not have access to data regarding the frequency of follow-up or details of interventions.

### 3.2. Participant Characteristics

The participant’s characteristics are described in Table 1. Children in the control group had higher Apgar scores; other birth characteristics were comparable between the groups. Within the group of children with a history of HIE, three were classified as mild encephalopathy (2/3 had abnormal post-TH MRI), six moderate (4/6 with abnormal post-TH MRI), and one severe (abnormal post-TH MRI) (Sarnat scores of 1, 2, and 3, respectively). There was some missing birth data, specifically, the delivery mode for one child.

Table 1 also includes clinical details from their birth hospitalization. Newborn MRIs were completed post-TH, on days 4–5 of life. Abnormalities included various areas of abnormal signals in the cerebral and periventricular white matter, basal ganglia (BG), and thalamus, and one infant displayed focal cystic changes in the anterior limb of the internal capsule, one with focal areas of ischemic infarct in the posterior limb of the internal capsule, the parietal and occipital lobe. EEGs were performed during TH and were considered abnormal if seizure activity was detected or if read by a neurologist as abnormal (background abnormalities). Seizures were included if detected by an EEG or amplitude-integrated EEG (Table 1). 

### 3.3. Brain Volumes

While the total brain volumes did not differ between the groups (Figure 2a), hippocampal volumes were smaller in children with HIE compared to the healthy controls by about 10%. This was true for both the left (*p* = 0.03) and right hippocampi (*p* = 0.01) and overall (*p* = 0.02) (Figure 2b). This difference remained when adjusting for the total intracranial volume (ICV) (*p* = 0.008 for the whole hippocampus). When the hippocampus was segmented into subregions, as was performed in other studies using Freesurfer [17,20], all were smaller for children with HIE than the controls (cornus ammonis fields *p* = 0.01, subiculum *p* = 0.04, dentate gyrus *p* = 0.006) (Figure 2c). Lastly, when the hippocampus was divided into head, body, and tail, these volumes were significantly smaller for the HIE group than for the controls (*p* = 0.03, *p* = 0.008, *p* = 0.03, respectively).

There was no significant difference in the size of the frontal, parietal, occipital, or temporal lobes between groups. Nor was there a group difference in the volume of other gray matter structures (thalamus, putamen, caudate nucleus, and amygdala). There was no difference in the size of the posterior hypothalamus, which contained mammillary bodies (which are associated with memory function) between the groups.

### 3.4. Developmental Testing

The median FSIQ was 109 (range 99–128) for the controls and 98.5 (range 79–126) for the HIE group, although this difference was not statistically significant (*p* = 0.075). The Global Executive Function (BRIEF-P) median for the controls was 48 (range 34–54) and 42 (range 33–76) for the HIE group (higher scores represent worse executive function). Aside from the Visual Spatial scores, which were significantly lower (*p* = 0.05) for infants with HIE, none of these subtest scores for the WPPSI, NEPSY-II, or BRIEF-P differed significantly between the groups (Table 2). 

In children with HIE, there was a moderately positive correlation between the hippocampal volume and sentence recall (NEPSY-II) (r = 0.66, *p* = 0.038) (Figure 3). Narrative memory (NEPSY-II) also had a moderate positive correlation; however, this only trended toward significance in this small sample (r = 0.5, *p* = 0.14). Both memory subscales retained a moderate positive correlation when correcting for ICV (r = 0.37 and r = 0.41, respectively); however, these correlations did not reach statistical significance. Within the control group, there was no relationship between the hippocampal volume and memory function (narrative memory r = −0.01, *p* = 0.98, sentence recall r = −0.22, *p* = 0.61).

### 3.5. Correlations between Neonatal and 5-Year Data

There was no relationship between newborn memory function as assessed by ERP results (using a slow wave voltage difference between the mother’s and stranger’s voices representing the strength of discrimination between the familiar and novel) [13] and hippocampal volumes at 5 years. This remained true when adjusting for the total ICV.

## 4. Discussion

Children with a history of HIE are at risk of decreased hippocampal volumes and lasting memory dysfunction, which negatively impacts school performance and daily functioning. Treatment using TH improves general outcomes in these infants [3,4,5]. This study was unique in that it showed *despite* treatment with TH and the preservation of the total brain volume, our cohort of children with HIE had smaller hippocampi than the healthy controls at 5 years, and smaller hippocampal volume was associated with lower memory function in children with a history of HIE.

Maneru et al. [10] compared teens with a history of term birth and HIE *without* TH to healthy controls at the age of 16 years and found hippocampal atrophy in those with HIE compared to the controls. Notably, the teens with a history of HIE were without neurologic deficits and had IQs greater than the population mean, not unlike our cohort. Annink et al. [8] showed that term-born children aged 9–10 years with a history of moderate HIE (*n* = 26) *without* TH had smaller hippocampi compared to the controls (12.6% smaller) and that the hippocampi of children with mild HIE (*n* = 26) were not significantly different from the moderates. Our study found similar results in children who received TH, including some with mild HIE; hippocampal volumes in those with a history of HIE were about 10% smaller than the controls.

Most recently, Annink et al. compared children with a history of term birth HIE who *did* vs. *did not* receive TH and showed both groups had reduced mammillary body (MB) size on MRI at 10 years of age. They further showed that hippocampal volumes were smaller in children with MB atrophy compared to those with normal mammillary body volumes [12]. Their study also showed a relationship between the MB volume, hippocampal volume, and memory function. Our results were in alignment with their findings and suggested that TH was not effective in completely mitigating injury to the hippocampus after HIE. While Freesurfer does not segment out the MB alone, we looked at the posterior hypothalamus, which included the MB, and did not find a relationship with the hippocampal volume. As other reports have shown the involvement of the MB after HIE, further studies evaluating the hippocampus and memory should include mammillary body measurements.

Based on the birth histories of our cohort of infants, we presumed that this group of study children experienced intrapartum hypoxic-ischemic events. The hippocampus is exquisitely vulnerable to hypoxic-ischemic injury secondary to oxidative stress and subsequent cellular damage [21]. Though TH has been shown to decrease death and disability after HIE and appears to improve global functioning, there seems to be limited effect on the hippocampus. While some animal studies have shown the attenuation of hippocampal injury using TH after hypoxia-ischemia (HI) [22,23], others have shown that the hippocampus is not protected by TH [24,25]. Further, previous clinical reports suggest that cooling may have more of an impact on the cortex and basal ganglia than the hippocampus, even on early imaging [26]. The present study supports the research that some injury persists and manifests as memory dysfunction and hippocampal atrophy even after TH. 

The hippocampus is at high risk for atrophy secondary to apoptosis and necrosis after a hypoxic-ischemic event due to its high metabolic activity and oxygen demand in a term fetus/infant. This is a consequence of decreased cellular energy reserves, reactive oxygen species, inflammation, and the disruption of calcium homeostasis associated with HI [21]. As HI leads to multiple processes that negatively impact the brain, it is likely that multiple processes contribute to the long-term memory dysfunction experienced after such an event as well. The calcium buffering protein, calbindin-1 (Calb-1), is at high levels in the developing hippocampus and is diminished after neonatal HI. The lack of calcium buffering could contribute to increased cell death and has not been attenuated by TH in animal studies [24]. Calb-1 levels have been associated with memory function in animal studies of neonatal HI and aging [24,27].

Another explanation for the lack of full attenuation of injury to the hippocampus is a delay in the injury of hippocampal interneurons compared to pyramidal cells, as suggested by Chavez-Valdez et al. [25]. Their mouse model of HIE shows the death of pyramidal cells 24 h after a hypoxic-ischemic event but a more delayed decrease in interneurons, as well as maturational and biochemical disruptions, even with TH. Furthermore, their data suggest that the loss of interneurons is not solely dependent on the degree of pyramidal cell injury, and subsequently, the rescue of pyramidal cells with TH does not produce an equal effect on interneurons. 

There is a need for focusing new treatments after HIE on hippocampal injury and repair. Kwak et al. showed that neural stem and progenitor cells (NSPCs), which are present in the dendrate gyrus and are important for postnatal development, hippocampal function and injury repair, are diminished after HI and partially rescued by TH in mice [22]. A promising therapy to further increase NSPCs after HI is treatment with sildenafil: a selective inhibitor of phosphodiesterase type-5. In an animal study by Yazdani et al., sildenafil increased NSPCs, decreased inflammation, and preserved hippocampal structure after HI through pathways targeting NSPCs [28]. 

Although it might be presumed that the inclusion of infants with mild HIE could be an explanation for some of the negative findings in this study, two of the three infants who scored as mild HIE had abnormal newborn MRI results, which suggested that the event leading to encephalopathy was still significant. Studies are increasingly showing that mild HIE has a significant impact on outcomes, and many centers offer TH to this group [6,29,30].

This study’s first limitation was a small sample size. Although recruiting from a previously studied group allowed for longitudinal analysis and is a strength of the study, the attrition rate was high and affected our sample size at 5 years. Further, many of the healthy control parents wanted their child to participate in the developmental testing, but not the MRI, for reasons that were not always verbalized. Because of this, we were unable to make correlations between some of the important newborn findings and the 5-year results (for example, correlating normal vs. abnormal newborn MRI and 5-year results). Though our 5-year data showed that developmental testing scores did not generally differ statistically between groups, there was a much wider range of scores in the HIE group with FSIQ trending lower than the controls; this may be more significant with a larger sample. Further, at least one child was too impaired to undergo testing, and thus, that level of disability was not accounted for in the testing scores, nor were we able to evaluate the brain volume. There may have been others from the original cohort that did not participate in the follow-up but were also more severely impaired despite comparable baseline characteristics between those who did and did not follow up at 5 years. However, by studying a cohort of high-functioning children, with a large average range of IQs, we were able to underscore that brain development and memory function was altered even after a mild injury. 

## 5. Conclusions

Our previous study of this cohort showed preserved recognition memory function, though altered circuitry, in the newborn period [13]. This may be an early marker of hippocampal and memory changes that persist into early childhood. This study showed smaller hippocampal volumes at 5 years compared to the controls, despite treatment with TH. Further evaluation with a larger sample should be conducted to confirm this finding and explore this correlation to the degree of initial encephalopathy and early MRI findings. Clinicians should be cautious when interpreting normal range generalized function testing during infancy and childhood. Despite the current treatment of HIE using TH, the hippocampus appears to remain at risk. Further, in addition to adverse impacts on childhood memory, there may be long-term effects related to premature aging and the degeneration of the hippocampus after an acute neonatal event [31]; thus, continued longitudinal studies are needed. Future studies that specifically target a reduction in injury to the hippocampus may improve the overall outcomes for children with neonatal HIE.

## Figures and Tables

**Figure 1 children-10-01005-f001:**
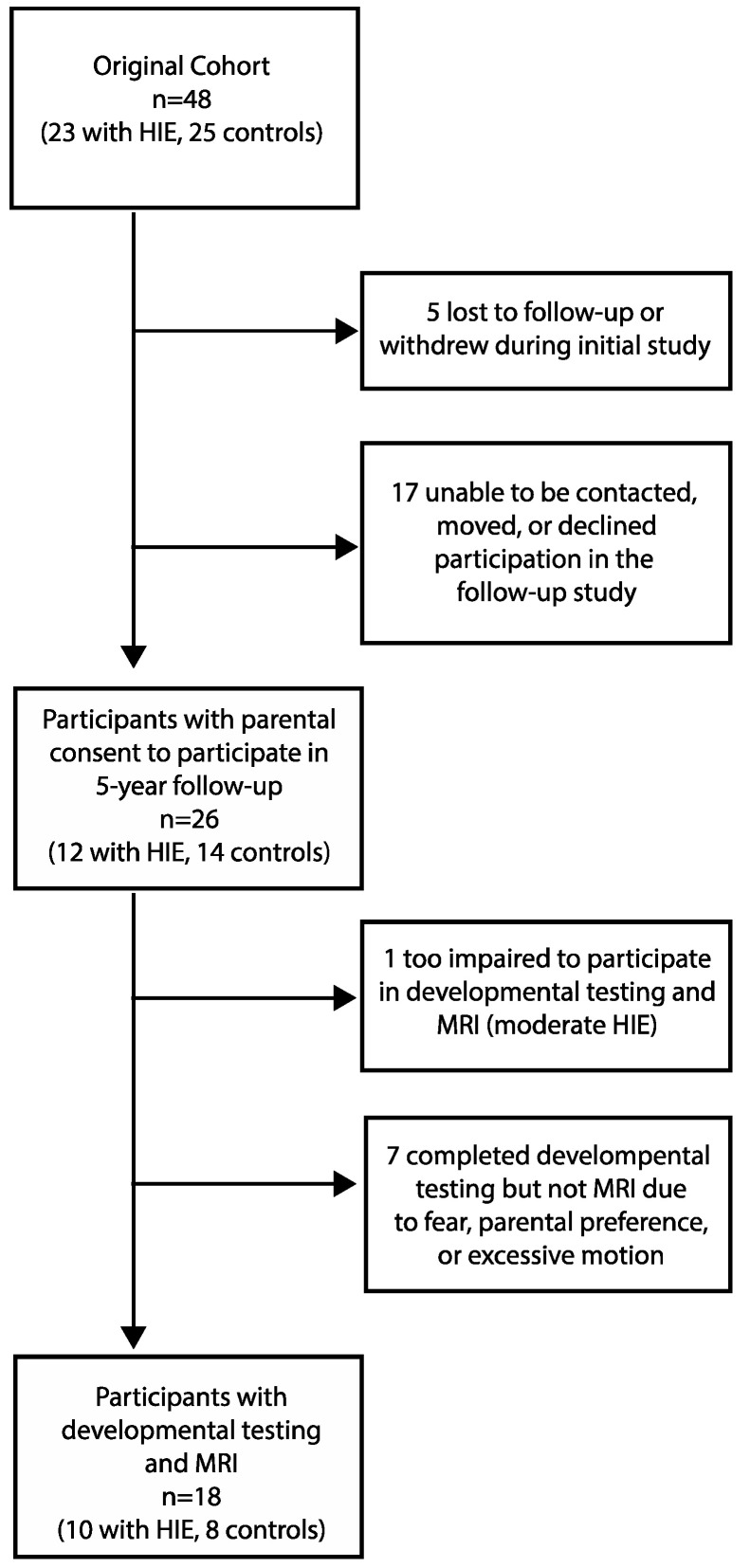
Flowchart of subject participation in the longitudinal study.

**Figure 2 children-10-01005-f002:**
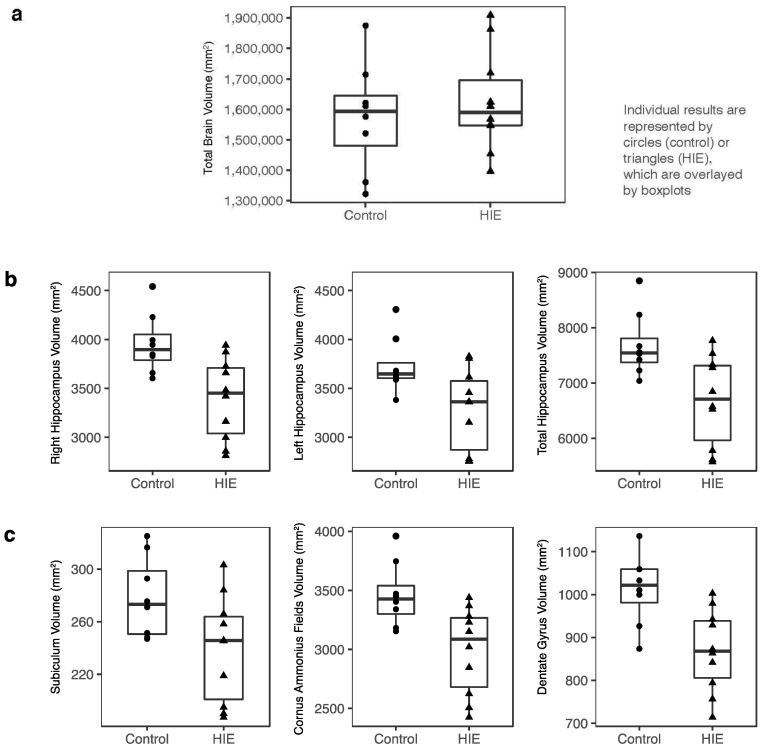
Brain volume comparisons between groups. Individual results are represented by dots or triangles, which are overlayed by boxplots. (**a**) Comparison of total brain volume at 5 years between control and HIE groups; there was no significant difference. (**b**) Hippocampal volumes divided into left, right, and bilateral hippocampus, and (**c**) subregions. The hippocampal volumes for the HIE group were significantly smaller than the control group in all subregions shown in (**b**,**c**) (*p* < 0.05).

**Figure 3 children-10-01005-f003:**
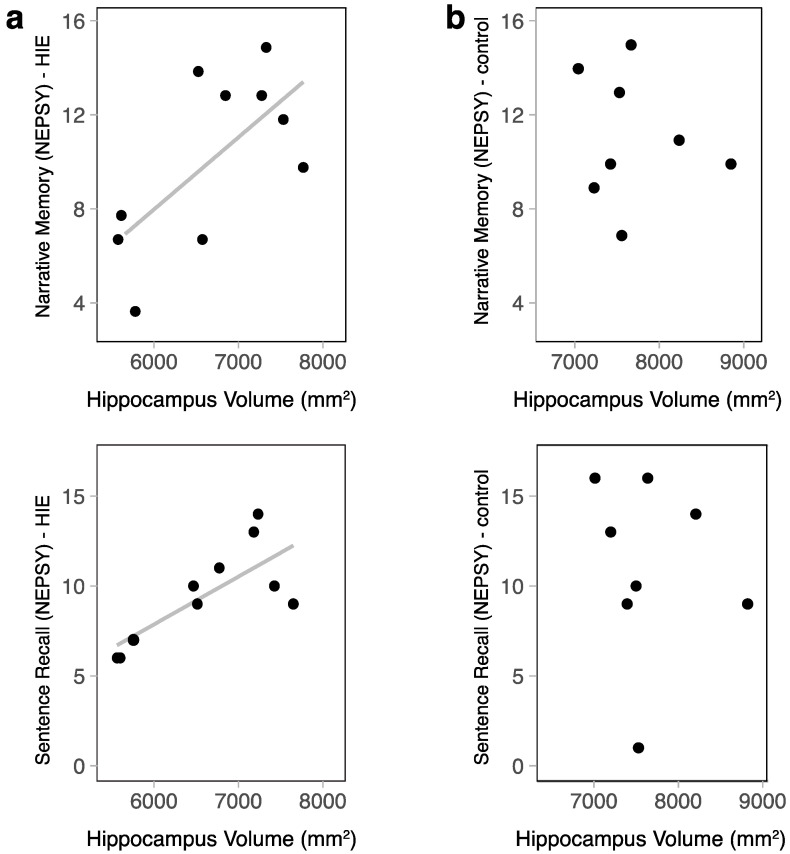
The relationship between hippocampal volume and memory function was assessed via a standardized psychometric assessment using NEPSY-II subtests. (**a**) In the HIE group, narrative memory scores increased as the hippocampal volume increased (r = 0.5, *p* = 0.14), and sentence recall scores increased as hippocampal volume increased (r = 0.66, *p* = 0.038). (**b**) In the control group, there was no relationship between hippocampal size and memory function.

**Table 1 children-10-01005-t001:** Participant characteristics.

Participant Characteristics	HIE (*n* = 10)	Control (*n* = 8)
Age at follow-up MRI, years	5.08 (5.02–5.42)	5.13 (4.96–5.58)
Birth gestational age, weeks	39.7 (39.0–40.5)	39.3 (38.9–39.6)
Birth weight, kg	3.69 (3.48–4.02)	3.6 (3.23–3.66)
Sex, Male	6 (60)	4 (50)
Cesarian section birth ^a^	3 (30)	4 (57)
Mother’s age at birth, years	31 (25–34)	33.5 (30.8–36.3)
Mother’s education, >HS	9 (90)	7 (87.5)
1 min Apgar score	1.5 (1–2) *	8 (7.75–8.25)
5 min Apgar score	4 (3–5) *	9 (9–9)
Sarnat score	2 (1.25–2)	
Initial pH ^b^	6.98 (6.92–7.05)	
Newborn MRI abnormal	7 (70)	
Newborn EEG abnormal	4 (44)	
Newborn seizures	3 (30)	

Data presented as median (IQR) for continuous data or number (%) for categorical data. HS = high school. ^a^ Missing data from one participant. ^b^ From umbilical cord blood gas or infant blood gas within 1 h of life. * Significant difference between groups, *p* < 0.05.

**Table 2 children-10-01005-t002:** Developmental testing scores.

Test	HIE (*n* = 10)		Control (*n* = 8)	
	Median	IQR	Median	IQR
**WPPSI**				
Full scale IQ	98.5	(84–106)	109	(101–114)
Visual spatial	94 *	(91–97)	109	(102–110)
Working memory	100	(90–108)	98.5	(94–104)
Processing speed	103	(87–106)	95.5	(91–97)
Fluid reasoning	97	(94–112)	111.5	(96–117)
Verbal comprehension	94.5	(84–109)	117	(113–123)
**NEPSY-II**				
Sentence recall	9.5	(8–11)	11.5	(9–15)
Narrative memory	11	(7–13)	10.5	(10–13)
Comprehension of instructions	11.5	(9–14)	14	(12–14)
**BRIEF-P**				
Global executive function	42	(35–49)	48	(42–52)
Inhibit	41	(37–49)	52.5	(44–56)
Shift	43	(42–54)	40	(40–44)
Plan/organize	43	(34–49)	48.5	(40–54)
Emotional control	43	(36–46)	46	(38–51)
Working memory	42	(38–58)	44	(41–50)

WPPSI average composite score 100 ± 15. NEPSY-II average scaled score 10 ± 2. BRIEF-P average T-scores 50 ± 10 with scores > 59 signifying worse executive function. * *p* = 0.05.

## Data Availability

The data presented in this study are available on request from the corresponding author. The data are not publicly available due to privacy restrictions.

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
