# Peer review of "Reduced Hippocampal Volumes in Children with History of Hypoxic Ischemic Encephalopathy after Therapeutic Hypothermia"

_children, 2023, doi:10.3390/children10061005_

Round 1

Reviewer 1 Report

Dear editor,

Thank you for taking the opportunity to review the article titled "Reduced Hippocampal Volumes in Children with History of Hypoxic Ischemic Encephalopathy After Therapeutic Hypothermia".

In this study, the relationship between brain MRI and hippocampal volumes and memory functions at 5 years of age in infants with HIE was investigated.

The study included very few patients and control groups. In addition, the number of patients with different HIE stages is very small. I am of the opinion that the investigation of the working hypothesis with such a small sample size will not have sufficient scientific power.

MRI scan times should be clearly stated in the study. MRI results taken after TH should be included: in which patients diffusion restriction was detected (mild-moderate-severe HIE?) ? It should be stated whether a developmental follow-up program was applied to the study group, developmental intervention-rehabilitation-etc. was given.

Sincerely.

Reviewer 2 Report

The manuscript present data indicating that therapeutic hypothermia applied in neonatal hypoxia-ischemia does not prevent specific brain changes, especially decrease in hippocampal volume. However, presented data do not indicate differences in specific brain function investigated by the authors. The question is, what exactly the authors wanted to show in their experiments. It is common knowledge that therapeutic hypothermia is not effective enough to prevent HI evoked changes in the brain. The authors should emphasized the aims of their study more clear.

The comparison of investigated parameters with  changes observed in children who did not receive hypothermic treatment would increase the value of presented results.

Other comments:

Page 3, lines 136-138 – how the authors found that there was no changes between group that appeared for tests 5 years later and those who did not come?

Page 4, line 149 – what “newborn period” means exactly?

Table 1 is not clear, what “Male” refers to and what mean numbers in brackets?

Page 5, line 168 – “hippocampus head, body and tail” – is it an equivalent for subiculum, dentate gyrus and cornus ammonis? It should be cleared.

Page 7, section 3.5 – why the authors compare memory functions assessed in newborns with hippocampi volume measured 5 years later?

The conclusions are not convincing and do not bring any new ideas.

Reviewer 3 Report

The paper entitled 'Reduced Hippocampal Volumes in Children with History of Hypoxic Ischemic Encephalopathy After Therapeutic Hypothermia' has good significance. This study on humans emphasizes that hypoxic encephalopathy after therapeutic hypothermia reduces hippocampal volume.  

I have comments for this work:

1) The experimental groups used in this study are poorly described in the Materials and Method section. 

2) Authors need to pay more attention to possible mechanisms involved in the loss of hippocampal volume: describe in more detail cellular and molecular mechanisms based on the literature. Other similar studies using animal models need to be discussed in the Discussion section in more detail.

Reviewer 4 Report

I read with interest your paper "Reduced Hippocampal Volumes in Children with History of Hypoxic Ischemic Encephalopathy After Therapeutic Hypothermia".

Although an important topic and finding, it is not a new observation. Annink et al has already described this in a larger cohort in cooled and non-cooled group. So nothing new and now even with a smaller group of patients. Also no discussion on the MB, which maybe a more sensitive early biomarker of damage to the limbic system. It looks like a better way of looking at memory problems in asphyxiated neonates, because if the have damage to the MB the have a very high chance of memory problems, which is less obvious in case of hippocampal volume loss. What is a cut-off volume loss of HC which gives a memory problem? Which areas of the HC should be affected? Subiculum?

In the discussion: While Freesurfer does not segment out the MB alone, we looked at the posterior hypothalamus, which includes the MB, and did not find a relationship with hippocampal volume (data not shown). This is of interest but hard to believe, knowing the connection between HC and MB. Maybe if a subdivision is made of the HC, parts are more atrophic than other parts and better correlated with MB. 

Was the posterior hypothalamus normal in all studied patients? 

Did the authors looked at the neonatal scans as well?

Would be of interest if HC damage was already visible at neonatal period or not? Can the HC volume loss be compensated by training or like in MB where the damage and atrophy will not change over time?

Good 

Round 2

Reviewer 1 Report

Dear authors,

I believe that this revised article can be published.

Sincerely.

Author Response

The reviewer was happy with the changes. No further edits for this reviewer.  Thank you.

Reviewer 2 Report

The authors responded to the comments in the review in a satisfactory manner.

Reviewer 4 Report

The revision made by the authors substantially improved the manuscript.

The only suprising outcome is still that the authors did not see damage to the MB on the neonal scans? and on FU. There are several reports showing that nearly 30% of neonates with HIE will have damage to the MB, which may be an easier biomarker to recognize on FU MR imaging. Maybe the small number of cases could partly explain this. The imaging protocol at 5 years is sufficient enough to assess atrophy of the MB and not only at the HC. Especially knowing that the MB are connected to the subiculum area.

Author Response

Thank you for your comments.  We have added the following sentence in the discussion at the end of the section where MB is discussed: "As other reports have shown involvement of the MB after HIE, further studies evaluating hippocampus and memory should include mammillary body measurement."